# Self-Supervised Learning via Flow-Guided Neural Operator on Time-Series Data

**Duy Nguyen**[*]                                                                       *dhnguyen@caltech.edu*
*Department of Computing and Mathematical Sciences, Caltech*

**Jiachen Yao**[*]                                                                      *jiachen.yao@caltech.edu*
*Department of Computing and Mathematical Sciences, Caltech*

**Jiayun Wang**[*]                                                                         *peterw@caltech.edu*
*Department of Computing and Mathematical Sciences, Caltech*
*School of Computational Science and Engineering, Georgia Tech*

**Julius Berner**                                                                         *jberner@nvidia.com*
*NVIDIA*

**Animashree Anandkumar**                                                                    *anima@caltech.edu*
*Department of Computing and Mathematical Sciences, Caltech*

**Reviewed on OpenReview:** *https://openreview.net/forum?id=YAYW9Y173z*

## Abstract

Self-supervised learning (SSL) is a powerful paradigm for learning from unlabeled time-series data. However, popular methods such as masked autoencoders (MAEs) rely on reconstructing inputs from a fixed, predetermined masking ratio. Instead of this static design, we propose treating the corruption level as a new degree of freedom for representation learning, enhancing flexibility and performance. To achieve this, we introduce the Flow-Guided Neural Operator (FGNO), a novel framework combining operator learning with flow matching for SSL training. FGNO learns the frequency functions represented through the Short-Time Fourier Transform (STFT). We extract a rich hierarchy of features by tapping into different network layers and flow times that apply varying strengths of noise to the input data. This enables the extraction of versatile representations, from low-level patterns to high-level global features, using a single model adaptable to specific tasks. Unlike prior generative SSL methods that use noisy inputs during inference, we propose using clean inputs for representation extraction while learning representations with noise; this eliminates randomness and boosts accuracy. We evaluate FGNO across three biomedical domains, where it consistently outperforms established baselines. Our method yields up to 39% relative AUROC gains over the MAE baseline in neural signal decoding (BrainTreeBank), 18% RMSE reductions in skin temperature prediction (DREAMT), and over 20% improvement in accuracy and macro-F1 on SleepEDF under low-data regimes. Furthermore, experiments under aggressive downsampling further show robustness to severe bandlimiting, highlighting FGNO's capability to learn in the function space. These results highlight FGNO's robustness to data scarcity and its superior capacity to learn expressive and adjustable representations for time series.

## 1 Introduction

Time-series data are common across domains such as healthcare (Johnson et al., 2016) and weather forecasting (Pathak et al., 2022). Learning useful supervised representations from temporal signals can be challenging when labels are scarce. Thus, self-supervised learning (SSL) has become a compelling technique, enabling

---

[*]Equal contribution

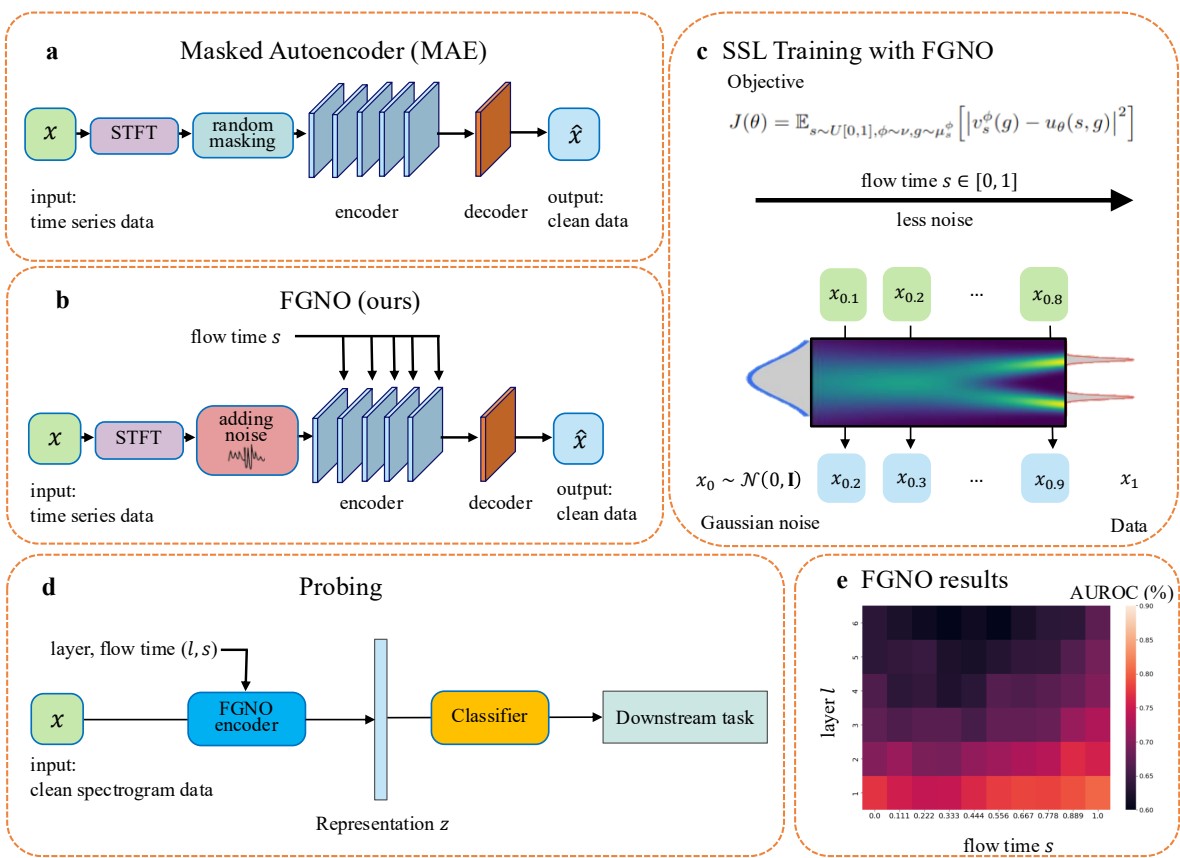

Figure 1: **(a)** A common self-supervised learning (SSL) baseline is Masked Autoencoder (MAE, He et al. (2021)), where the input data is randomly masked at a *fixed* ratio and then fed to encoders and decoders to reconstruct the clean data. MAE learns useful representations by inpainting the missing part. **(b)** We ask if the ratio can *vary* continuously and propose flow-guided neural operator (FGNO), which is based on flow matching that progressively transforms noisy inputs, corrupted at flow time $s$, to clean data by predicting intermediate velocities. Both methods first transform the time series data to spectrograms via STFT (short-time Fourier transform) to extract local time-frequency features. **(c)** FGNO is pre-trained in a self-supervised manner using the flow-matching objective. FGNO performs flow matching between distributions over time-frequency functions, using a Gaussian-process prior. Under downsampled evaluation, FGNO is empirically robust to severe bandlimiting. The decoder shown is a shallow spectrogram reconstruction head used solely during pretraining and discarded for downstream tasks. **(d)** After SSL pretraining, representations are probed by training a small classifier for downstream tasks. Compared with existing generative SSL methods, we use clean input data instead of noisy data as input and achieve similar performance with no randomness from the noise generation. **(e)** FGNO's performance on sleep/wake classification from the DREAMT dataset (Wang et al., 2024c). A single FGNO produces flexible representations with various layer and flow time $(l, s)$. The top-performing $(l, s)$ pairs form a continuous, structured region in the hyperparameter space.

models to exploit large collections of unlabeled time series data. Prior work adapts ideas from natural language processing and computer vision, such as BERT (Devlin et al., 2019), masked autoencoders (MAE) (He et al., 2021), and contrastive objectives (Siméoni et al., 2025). Recently, increasingly capable time-series foundation models (Ansari et al., 2024) and flow-based generative models (Zhang et al., 2025) have gained significant attention. Although focusing on forecasting tasks, their SSL capability is of considerable interest.

Despite the progress of self-supervised learning in time-series modeling, learning generalizable representations remains challenging due to the heterogeneous nature of real-world data and the diversity of downstream tasks. Time-series signals are often recorded at different sampling rates, and standardizing them through upsampling or downsampling distorts their intrinsic characteristics. In the DREAMT dataset (Eldele et al.,

2024), for instance, wearable device signals are collected at multiple frequencies ranging from 4 Hz to 200 Hz. Aligning such data requires interpolation and resampling that risk blurring fine-grained events, such as micro-arousals or transient heart rate variability patterns, thereby contaminating the learned representation.

In addition to resolution mismatches, downstream tasks often demand representations at different temporal and semantic scales. Sleep-stage classification relies on local patterns in the length of seconds, whereas apnea-hypopnea index (AHI) regression requires integrating information across an entire night. Similar multi-scale demands also arise in other domains: clinical forecasting on MIMIC-III (Johnson et al., 2016) depends on long-term trends in vitals, whereas arrhythmia detection from electrocardiogram (ECG) relies on millisecond-level waveforms. Despite such needs, most SSL approaches are optimized for a fixed pretraining objective and yield a single latent representation, limiting their adaptability across tasks with different context length and frequency dependency.

These considerations motivate *a unified pretraining framework that preserves the fidelity of multi-resolution time-series signals while yielding flexible, task-adjustable representations.*

Neural operators learn mappings directly in the function space of signals, offering a natural framework for time-series modeling where data can be viewed as functions over time (Azizzadenesheli et al., 2024). This approach has achieved state-of-the-art results across various time-series domains, including forecasting, imputation, and anomaly detection (Li & Yang, 2023). Often, they operate directly on raw temporal signals. STFT provides a task-aligned representation for biomedical and wearable signals, in which discriminative patterns frequently appear as localized events or activity within specific frequency bands. When operating on raw one-dimensional signals, a model must implicitly disentangle this localized time-frequency structure from the mixed waveform. STFT explicitly exposes both temporal locality and frequency content, while retaining the evolution of spectral patterns over time. This representation is especially useful for signals such as EEG, BVP, and accelerometry, whose downstream labels can depend on oscillations, frequency-band activity, or short duration physiological events.

To adapt to tasks of different abstract levels, generative modeling offers a complementary perspective. Diffusion- and flow-based methods (Ho et al., 2020; Lipman et al., 2023) learn to map simple noise distributions to complex data distributions and are trained with self-supervised signals. Recent studies on images suggest that internal representations taken at different noise levels naturally organize from low-level textures to high-level global features, providing a *continuous* control knob for multi-scale features (Tang et al., 2023). A concern with these methods, however, is that their inputs are noised at different noise levels, potentially leading to information loss during downstream tasks.

We propose the **Flow-Guided Neural Operator** (FGNO; Fig. 1), a self-supervised framework that pretrains a *flow matching* model on spectrograms of unlabelled data and extracts task-specific representations by selecting a network layer $l$ and flow time $s$ (a higher $s$ corresponds to a lower noise level $\sigma_s$). We leverage the Short-Time Fourier Transform (STFT) to embed 1D signals into time-frequency functions that expose both temporal locality and frequency-band content. We then define the flow-matching path using a Gaussian-process prior, rather than independently sampled Gaussian noise, so that the initial state has structured covariance over the time-frequency domain. This construction allows a function-space learning during the flow matching process. Empirically, the resulting representations remain robust when high-frequency information is removed through severe downsampling.

We treat features $\phi_{l,s}(x)$ (the hidden states at layer $l$ conditioned on time $s$ for clean input $x$) as a hierarchy of representations. After pretraining, we train a probing head (classifier) on top of $\phi_{l,s}$ while freezing the backbone. This design turns the flow time $s$ into a practical, tunable degree of freedom that allows users to emphasize fine temporal detail (higher $s$, shallower $l$) or higher-level semantics (lower $s$, deeper $l$) with a *single* pretrained model. Unlike prior generative SSL methods that use noisy inputs during inference, which may contaminate information, FGNO uses clean inputs for representation extraction. This enables superior performance with no randomness.

We observe that the optimal choice of network layer $l$ and flow time $s$ is task-dependent: tasks requiring precise local timing benefit from lower noise and earlier layers, whereas tasks relying on global context prefer higher noise and deeper layers. Selecting $(l, s)$ per task yields consistent gains over MAE- and contrastive-based baselines, including up to 39% relative AUROC improvements over the MAE baseline on neural signal

decoding (BrainTreeBank, Wang et al. (2024a)), 18% RMSE reductions on skin temperature regression (DREAMT, Wang et al. (2024c)). Moreover, our approach demonstrates exceptional robustness to data scarcity: on SleepEDF, FGNO maintains 93.5% accuracy and 89.0% macro-F1 with only 5% labeled data, representing over 20% improvements compared to strong baselines; on Epilepsy, FGNO achieves 94.1% accuracy and 90.3% macro-F1 under the same setting, effectively matching the full-data results.

In summary, our contributions are as follows:

- **An SSL framework combining flow matching and operator learning for time series.** We pretrain flow matching models on time-frequency representations of 1D signals, generated via STFT. This enables training at one resolution while generalizing to other resolutions during downstream tasks, with minimal performance degradation.

- **Flow time as a mean to control features.** We demonstrate that flow time $s$ provides an explicit and practical control over representation granularity, yielding a rich, multi-level feature hierarchy by varying the flow time $s$ and network layer $l$.

- **Performance gains with clean-input representations.** During downstream probing, we extract representations from clean inputs rather than generating noisy inputs as commonly done when probing time-conditioned models. Our ablation shows that clean-input extraction achieves comparable or better performance while eliminating variation caused by inference-time noise.

- **Empirical advantage on biomedical benchmarks.** FGNO significantly outperforms established baselines across diverse tasks, including up to 39% relative AUROC improvement over the MAE baseline in neural signal decoding, 18% RMSE reductions in skin temperature regression, and strong robustness under data scarcity on SleepEDF and Epilepsy—maintaining near full-data performance with only 5% labeled data.

## 2 Related Work

**Self-Supervised Learning (SSL)** SSL has shown great success in learning representations for various downstream tasks without labels, with applications to different modalities such as images (He et al., 2021; Wang et al., 2024b), audio (Gong et al., 2022), and videos (Tong et al., 2022; Gupta et al., 2024). Masked autoencoding (MAE) is a dominant self-supervised paradigm. For instance, BrainBERT (Wang et al., 2023) successfully applies this technique to neural time-series data by training on masked spectrogram representations. Advanced SSL methods for time-series data further enhance representation performance. Contrastive Predictive Coding (CPC, van den Oord et al. (2019)) uses autoregressive predictions and contrastive losses to maximize mutual information in sequential data such as physiological signals. TS-TCC (Eldele et al., 2021) employs temporal-contextual contrasts and augmentations like jittering to improve robustness and generalization. We will compare with these methods in Section 4.

**Generative Models for SSL** Generative models, particularly diffusion and flow matching models, serve as powerful self-supervised learners, as their denoising objective inherently learns rich, multi-level data representations (Fuest et al., 2024). Existing work explores generative models for representation for visual data like images and videos (Fuest et al., 2024; Luo et al., 2023; Vélez et al., 2025; Tang et al., 2023). A dominant technique is to leverage intermediate activations from a pre-trained model's internal layers at various corruption timesteps, creating a feature hierarchy that spans from low-level textures to high-level global features. These extracted features are then used to train lightweight heads for downstream tasks like classification and segmentation. Our work builds on this foundation but shifts the focus from generation to representation learning. We investigate whether the flow matching objective can serve as a powerful self-supervised learning tool for multi-scale features of time-series data on downstream discriminative tasks.

**Representations from Diffusion Models** Recent research demonstrates that generative models implicitly learn hierarchically structured representations, ranging from low-level textures to high-level semantics, to solve the generation task. Earlier work such as SODA (Hudson et al., 2023) introduced an information bottleneck between a clean image encoder and a diffusion decoder, encouraging the encoder to learn compact

semantic representations. Recent methods such as REPA (Yu et al., 2025) and REG (Gao et al., 2025) improved diffusion generation through representation alignment and gradient-guidance refinement, respectively. CleanDIFT (Stracke et al., 2025) proves that generative priors can be decoupled from stochastic noise for robust discriminative tasks. Aligning with this "clean input" hypothesis, our work departs from stochastic probing. We design the flow matching objective to expose a continuous, controllable representation hierarchy that can be queried deterministically without the variance introduced by noise corruption.

**Neural Operators** Neural operators (Azizzadenesheli et al., 2024; Kovachki et al., 2023) are deep learning architectures specifically designed to learn mappings between infinite-dimensional function spaces. Neural operators have empirically achieved good performance for approximating the numerical solutions to partial differential equations (PDEs) (Kovachki et al., 2023; Li et al., 2024) and real-world applications such as computational imaging (Jatyani et al., 2025; Wang et al., 2025a;b). A prominent example is the Fourier Neural Operator (FNO, Li et al. (2020)), which uses Fourier-domain parameterization to model global dependencies.

## 3 Method

In this section, we introduce the Flow-Guided Neural Operator (FGNO), a novel framework for self-supervised representation learning. FGNO leverages the flow matching paradigm (Lipman et al., 2023) to learn generalizable representations. By constructing mappings in function space via Fourier-based representations (i.e., spectrograms), it functions as a neural operator capable of generalizing across resolutions.

The methodology has two stages: pre-training with flow matching, and probing with representation selection.

### 3.1 Self-Supervised Pre-training

**Data Embedding** The pre-training begins with embedding raw 1D signals into a time-frequency representation suitable for functional-space learning. To this end, we apply the Short-Time Fourier Transform (STFT) to convert the input signals $x \in \mathbb{R}^T$ into spectrograms $f \in \mathbb{C}^{N_f \times N_t}$, where $N_f$ denotes the number of frequency bins and $N_t$ the number of time frames. The STFT is defined as

$$\Phi(\tau, \omega) = \int_{-\infty}^{\infty} x(t)w(t-\tau)e^{-j\omega t}dt, \tag{1}$$

with $w(\cdot)$ as a sliding window function (e.g., Hann window) of length $W$, hop size $H$, and $\tau, \omega$ indexing time and frequency. We compute the magnitude spectrogram $\phi = |\Phi|$ as input to our model, which captures both temporal evolution and local patterns. STFT spectrograms are standard for preprocessing in speech recognition and audio analysis (Bäckström et al., 2022) but less common in SSL literature. This embedding provides a common time-frequency representation that can be aligned across sampling resolutions.

**Self-Supervised Learning via Flow Matching** Once embedded into magnitude spectrograms, the data functions $\phi \in \mathbb{R}^{N_f \times N_t}$ (sampled from the data distribution $\nu$) are used to pretrain a time-conditioned neural network $u_\theta(s, g) : [0, 1] \times \mathbb{R}^{N_f \times N_t} \to \mathbb{R}^{N_f \times N_t}$ via flow matching (Lipman et al., 2023). Flow matching provides a simulation-free objective for learning continuous transports that map a simple prior distribution (e.g., Gaussian noise) to the complex data distribution $\nu$. In our self-supervised setup, this objective implicitly learns representations by regressing toward a target vector field that guides the denoising of corrupted inputs.

Importantly, our approach is not an instance of Kerrigan et al. (2023)'s Function-space Flow Matching (FFM). FFM formulates its vector field directly on an infinite-dimensional function space. FGNO works on discrete STFT spectrograms. Moreover, our objective is to learn a time-conditioned backbone for extracting representations at different layers and flow times, rather than to construct a generative model for sampling function-valued observations that FFM focuses.

Specifically, for a given timestep $s \sim \mathcal{U}[0, 1]$, we construct a noisy interpolation $g \sim \mu_s^\phi$ between the clean data function $\phi$ and the noise distribution $\pi = \mathcal{N}(0, C_0)$, where $C_0$ denotes the covariance matrix induced by the Gaussian-process prior on the time–frequency grid, as

$$g = s\phi + \sigma_s\epsilon, \qquad \epsilon \sim \pi, \tag{2}$$

where $\mu_s^\phi$ is the conditional distribution of $g$ given $\phi$ and $s$. Here, $\sigma_s : [0,1] \to \mathbb{R}_+$ is a monotonically decreasing noise schedule from 1 to 0 that controls the noise level at $s$. We approximate the conditional expectation of the vector field $v_s^\phi(g)$ by training the model $u_\theta(s, g)$

$$J(\theta) = \mathbb{E}_{s\sim\mathcal{U}[0,1],\phi\sim\nu,g\sim\mu_s^\phi}\left[\left\|v_s^\phi(g) - u_\theta(s, g)\right\|_F^2\right], \tag{3}$$

with the target field given by

$$v_s^\phi(g) = \frac{\dot{\sigma}_s}{\sigma_s}(g - s\phi) + \phi, \tag{4}$$

where $\dot{\sigma}_s$ denotes the derivative with respect to $s$. Minimizing $J(\theta)$ equips $u_\theta$ with the ability to simulate the flow ODE

$$\frac{dg}{ds} = u_\theta(s, g) \tag{5}$$

that transports noise to data. This setup ensures self-supervision as the objective solely uses unlabeled data.

We numerically parameterize the velocity operator $u_\theta$ using a Transformer on the discretized STFT grid. Flow time $s$ is supplied through a sinusoidal embedding. The Transformer maps an evaluated time-frequency function to an evaluated velocity function of the same shape. The resulting pretrained model encodes multi-scale dynamics, from coarse structures at low $s$ (high corruption) to local details at high $s$ (low corruption), providing a versatile backbone for downstream probing.

**Residual operator parameterization.** The Transformer backbone is composed of residual blocks with skip connections. Denoting the hidden function representation at layer $l$ by $h^{(l)}$, each block has the form

$$h^{(l+1)} = h^{(l)} + \mathcal{T}_\theta^{(l)}\left(h^{(l)}, s\right), \tag{6}$$

where $\mathcal{T}_\theta^{(l)}$ denotes the time-conditioned attention and feed-forward transformation at layer $l$. $h^{(l)}$ and $h^{(l+1)}$ are discretized evaluations of functions on the time–frequency domain, while each residual block learns an incremental function-to-function update. The skip connections preserve the input function while the learned residual term modifies its representation, which is consistent with the residual operator blocks commonly used in neural operator architectures Kovachki et al. (2023).

### 3.2 Feature Extraction and Probing

**Feature Extraction with Clean Data** After pretraining, we freeze the Transformer weights $u_\theta$ and use the model as a feature extractor for downstream tasks. A key challenge here is the distributional shift between the noisy inputs $g$ encountered during pretraining and the clean samples available for fine-tuning. Typical generative SSL approaches address this by generating noisy inputs $g$ at a fixed or sampled timestep $s$ during inference, which introduces randomness and potential information loss from clean downstream data.

To avoid stochastic variation during downstream evaluation, we extract all candidate representations from clean spectrograms while varying only the selected network layer and flow time. Specifically, we provide the clean spectrogram $\phi$ as input and condition the frozen network on $s$ through its time embedding, without inputting a noisy sample. This deterministic procedure ensures that comparisons among candidate $(l, s)$ representations are not confounded by independently sampled noise and avoids the additional computational cost of noise generation.

Training on noisy interpolated states and probing on clean inputs introduces a difference between the pretraining and downstream input distributions. The lightweight probing head provides supervised adaptation to each downstream task. Although it cannot eliminate the distribution shift, the clean-versus-noisy ablation in Section 4.5 evaluates the practical effect of the two representation-extraction procedures and shows that clean-input probing provides competitive or better performance with deterministic features.

Formally, for a clean input spectrogram $\phi$ and desired flow time $s \in [0, 1]$, the representation at layer $l$ is obtained as

$$z_{l,s}(\phi) = u_\theta^{(l)}(s, \phi), \tag{7}$$

where $u_\theta^{(l)}$ denotes the activations at layer $l$ of the frozen model.

**Representation Selection and Probing**   The final stage of the FGNO involves training a lightweight probing head (e.g., a linear classifier or regressor) atop selected representations $z_{l,s}(\phi)$, using labeled data while keeping the backbone $u_\theta$ frozen.

Given that the model captures multiple representations, selecting the optimal representation configuration requires an evaluation of features at various layers $l$ and times $s$. Let $\mathcal{I}$ denote the set of candidate layer indices and $\mathcal{S}$ the set of candidate flow times. For each $(l, s) \in \mathcal{I} \times \mathcal{S}$, we train probe parameters $\psi$ on the training set while keeping the backbone frozen:

$$\psi_{l,s}^* = \arg \min_\psi \mathcal{L}_{\text{train}}(\psi; l, s). \tag{8}$$

We then select the representation whose trained probe minimizes validation loss:

$$(l^*, s^*) = \arg \min_{l \in \mathcal{I}, \, s \in \mathcal{S}} \mathcal{L}_{\text{val}}(\psi_{l,s}^*; l, s). \tag{9}$$

This selection process unlocks FGNO's full potential, allowing users to tailor feature granularity, balancing both temporal and global features without retraining the pretrained model.

## 4 Experimental Results

### 4.1 Datasets and Setup

**DREAMT** (Wang et al., 2024c) contains synchronized smartwatch and clinical-grade polysomnography (PSG) data from 100 participants, many with sleep disorders. For FGNO, a single model is pre-trained on the smartwatch's Blood Volume Pulse (BVP) and accelerometer (ACC) signals. This model's features are then evaluated on held-out participants for *two downstream tasks*: a binary sleep/wake classification and a skin temperature regression.

**BrainTreeBank** (Wang et al., 2024a) is a large-scale dataset of intracranial neural responses from 10 subjects watching Hollywood movies (43 hours in total). The dataset includes extensive linguistic annotations of the movie audio, such as transcripts and word onsets. Using a held-out set of subjects for probing, we evaluate our model on a binary speech presence classification task.

**Epileptic Seizure Recognition** (Andrzejak et al., 2001) consists of EEG recordings from 500 subjects, each with 23.6s of brain activity. The original dataset contains five classes, four of which correspond to non-seizure brain states. To focus more on seizure detection, we merge the four non-seizure classes into one and treat this as a binary classification problem (seizure v.s. non-seizure).

**SleepEDF** (Goldberger et al., 2000) is a widely used whole-night PSG sleep dataset from PhysioBank. We use a single EEG channel (Fpz–Cz) sampled at 100 Hz and classify each 30-second epoch into five classes: Wake, Non-REM sleep stages N1, N2, N3, and Rapid Eye Movement (REM) sleep.

For all datasets, we follow a strict chronological and subject-based splitting to prevent data leakage. The validation set is kept completely separate from the test set, and all model tuning is performed exclusively on the validation set. Specific preprocessing pipelines, STFT parameters, and detailed data split ratios are provided in Appendix A.

### 4.2 Sleep Classification and Skin Temperature Prediction on DREAMT

**Comparison to baselines**   From Table 1, FGNO outperforms the baselines on both sleep classification and skin temperature regression, with a narrow margin on classification and a substantial margin on regression.

Table 1: Performance comparison on downstream tasks from the DREAMT datasets. Our method (FGNO) achieves superior results on both classification (AUROC) and regression (RMSE) benchmarks compared to baselines.

| Task | Metric | FGNO (Ours) | MAE | Chronos |
|---|---|---|---|---|
| Binary Sleep Classification | AUROC (%) ↑ | **96.5** | 95.8 | 96.4 |
| Skin Temperature Regression | RMSE (°C) ↓ | **0.600** | 0.735 | 0.954 |

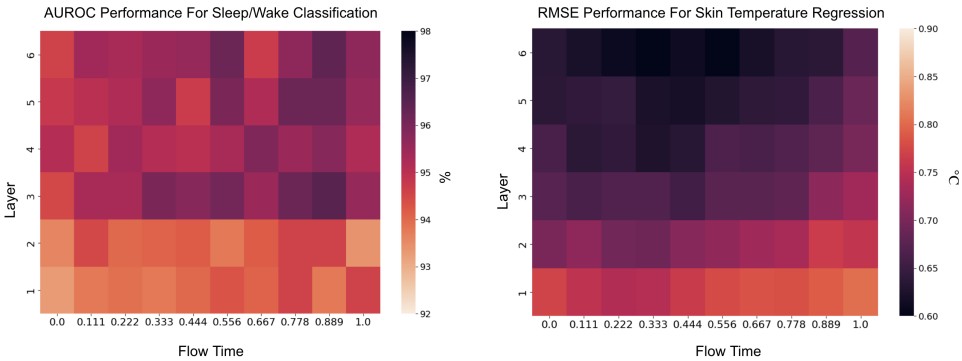

Figure 2: FGNO's performance across different layers and flow times on the DREAMT dataset. **Left:** Sleep classification AUROC (↑). **Right:** Skin temperature regression RMSE (↓). Darker is better.

It yields better AUROC compared to the MAE baseline across the optimal layers. Our peak score (96.5%) also surpasses the gradient boosting approach (92.6%) reported in DREAMT. Notably, our model achieved this using only raw 1D data, whereas the DREAMT baseline required additional clinical metadata (Apnea severity score) (Wang et al., 2024c), highlighting the capability of the self-supervised approach. For skin temperature regression, our best RMSE substantially improves upon the MAE baseline (0.735°C). Overall, the results highlight FGNO's ability to leverage both layer and flow time for highly predictive representations, whereas MAE is constrained to layer selection alone.

**Comparison to a foundation model**   Time-series foundation models excel in forecasting but may hold potential for SSL. Thus, we benchmark FGNO against Chronos (Ansari et al., 2024), a T5-based family of pretrained models that tokenizes raw signals for autoregressive training. For fairness, we selected a Chronos variant matching FGNO's parameter count and evaluated it as a feature extractor using two strategies: last-token hidden states or average pooling over all states. Average pooling yields Chronos's best results (96.4% AUROC for sleep classification; 0.954°C RMSE for regression), yet FGNO outperforms it narrowly on classification and substantially on regression (37% improvement). This underscores FGNO's role in data-efficient SSL for diverse downstream tasks.

**Insights from different layers and flow times**   As shown in Figure 2, sleep classification accuracy improves substantially in deeper layers. The best AUROC is achieved at layer 3 with low noise ($s = 0.89$). In contrast, skin temperature regression also favors deeper layers but achieves its lowest RMSE at moderate noise levels ($s \in [0.22, 0.56]$). We observe a trend that different levels of abstraction require different flow times. From Figure 6, it is clear that the physiological regression task depends more on global features, which align with lower to intermediate flow times (higher corruption levels). Conversely, classification tasks require more local patterns that prefer higher flow times (lower corruption), where temporal features are preserved. This allows us to leverage our understanding of the task to determine whether higher or lower flow times are needed, and reciprocally, to use the flow time to gain deeper insights into the task itself.

### 4.3 Classification Tasks on BrainTreeBank

To further benchmark FGNO against state-of-the-art methods, we compared our model against Brain-BERTWang et al. (2023), PopT Chau et al. (2024), and the DeepNN baseline on the BrainTreeBank dataset. As shown in Figure 3, FGNO outperforms BrainBERT and the DeepNN baseline on all four tasks despite being significantly smaller (370K parameters vs. 20M+ for baselines). Notably, while PopT incorporates explicit domain knowledge regarding electrode relationships, FGNO matches or exceeds it on three of the four tasks—comparable on Speech and higher on Volume and Pitch—purely through flow matching representation learning, trailing only on Sentence.

### 4.4 Robustness Under Data Scarcity

Medical data are often costly and limited. To evaluate FGNO's performance in data-scarce scenarios, we designed an experiment where the pre-training phase utilizes most of the available data without labels, after which the downstream probing head was trained on only 5% of the available labeled data and tested on held-out data. As shown in Table 2, FGNO achieves strong performance on both SleepEDF and Epilepsy even when only 5% of labeled data is available. On SleepEDF, our model maintains an accuracy of 93.5% and a macro-F1 of 89.0%, which is nearly identical to the performance obtained with 100% of the labeled data (93.9% ACC, 89.1% MF1) and outperforms all baselines by a large margin. On Epilepsy, FGNO achieves 94.1% accuracy and 90.3% MF1 under the 5% setting—matching its own full-data results and attaining the best accuracy among all methods, while remaining competitive with the strongest baselines (e.g., TS-TCC) in macro-F1.

We extended this evaluation to the DREAMT dataset. Table 3 demonstrates that even with only 5% of labeled data, FGNO retains competitive performance. For sleep classification, AUROC drops only slightly from 96.5% to 95.4%, and for skin temperature regression, the model maintains an RMSE of 0.710, outperforming the full-data baseline of MAE/BrainBERT (0.735).

These findings highlight the sample efficiency of our approach and suggest that FGNO is particularly well-suited for real-world biomedical applications, where large-scale labeled datasets are often scarce.

### 4.5 Ablation Study and Analysis

**Clean vs noisy input for probing**   A key component of our probing framework involves generating a noisy sample for a given clean input spectrogram $\phi$ and a desired flow time $s \in [0, 1]$:

$$g_s = s\phi + \sigma_s\epsilon, \quad \text{where } \epsilon \sim \mathcal{N}(0, C_0).$$

While effective, this step introduces computational overhead during inference. Moreover, its reliance on a random noise vector $\epsilon$ leads to unstable outputs. To investigate a more efficient alternative, we conducted an ablation study where we bypassed noise generation entirely. Instead of feeding the model a noisy input $g_s$, we provided the clean spectrogram $\phi$ directly and supplied the flow time value $s$ as an additional conditional embedding (the "Clean Input" method).

Our results, illustrated in Figure 4, show that the Clean Input method yields nearly identical mean performance to the Noisy Input method. For example, at layer 3 and time $s \approx 0.89$, the Clean Input method achieves a maximum score of 96.40% while the Noisy Input method yields 95.86%. This confirms the model learns to interpret $s$ as the corruption level. We also found that the Noisy Input method is sensitive to the random noise vector $\epsilon$, exhibiting performance variance across runs (std of 0.0039 at the same point). In contrast, the Clean Input method is entirely deterministic. This demonstrates that the clean approach is superior, as it is both more computationally efficient and provides a more stable, reliable inference pathway.

**Robustness to severe bandlimiting**   Our downsampling experiment evaluates robustness to missing high-frequency content rather than general discretization invariance. We pretrain FGNO once on BrainTreeBank signals sampled at 2048 Hz. At evaluation time, we downsample the raw signals before computing the STFT, with downsampling factors of $4\times$, $8\times$, $12\times$, $24\times$, $36\times$, and $48\times$. Downsampling reduces the Nyquist frequency

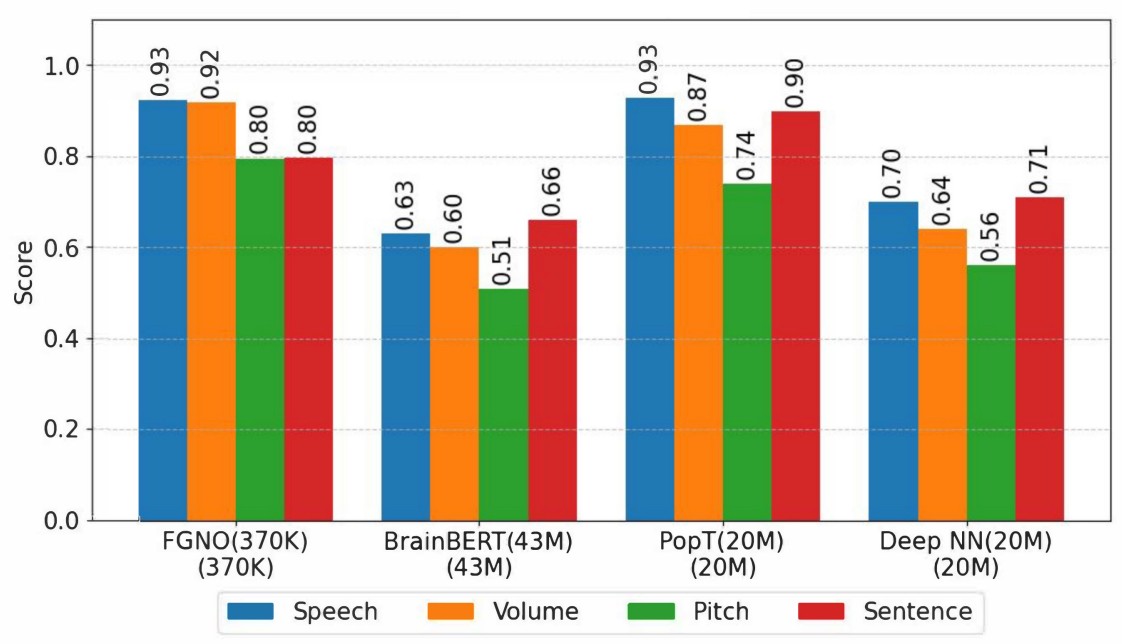

Figure 3: Model comparison in terms of size and AUROC on all four tasks of BrainTreeBank. The DeepNN's performance is quoted from Chau et al. (2024). FGNO outperforms latest baselines in most tasks while being significantly smaller in size.

Table 2: Accuracy (ACC) and macro F1-score (MF1) for all models evaluated with both 100% and a scarce 5% of labeled training data. FGNO consistently outperforms other methods, demonstrating its robustness and data efficiency in low-label regimes. **Bold** values indicate the best performance in each column. The baseline results are cited from Eldele et al. (2021).

| Baseline | 100% of labeled data | | | | 5% of labeled data | | | |
| | SleepEDF | | Epilepsy | | SleepEDF | | Epilepsy | |
| | ACC | MF1 | ACC | MF1 | ACC | MF1 | ACC | MF1 |
|---|---|---|---|---|---|---|---|---|
| Random Initialization | 35.6 | 23.8 | 90.3 | 81.1 | 22.8 | 22.8 | 75.5 | 70.5 |
| Supervised | 83.4 | 74.8 | 96.7 | 94.5 | 60.5 | 54.8 | 83.4 | 80.4 |
| SSL-ECG (Sarkar & Etemad, 2022) | 74.6 | 65.4 | 93.7 | 95.1 | 73.4 | 65.3 | 92.8 | 89.0 |
| CPC (van den Oord et al., 2019) | 82.8 | 73.9 | 96.4 | 94.4 | 76.3 | 70.5 | 90.2 | 90.2 |
| SimCLR (Chen et al., 2020) | 78.9 | 68.6 | 96.1 | 93.5 | 64.2 | 61.9 | 91.3 | 89.1 |
| TS-TCC (Eldele et al., 2021) | 83.0 | 73.5 | **97.2** | **95.5** | 77.0 | 70.9 | 93.1 | **93.7** |
| **FGNO (ours)** | **93.9** | **89.1** | 94.8 | 90.3 | **93.5** | **89.0** | **94.1** | 90.3 |

Table 3: Performance comparison on the DREAMT dataset for Skin Temperature Regression (RMSE) and Sleep Classification (AUROC). FGNO retains competitive performance even when trained with only 5% of labeled data, outperforming the full-data baseline in regression tasks.

| Baseline | 100% of labeled data | | 5% of labeled data | |
| | RMSE ↓ | AUROC ↑ | RMSE ↓ | AUROC ↑ |
|---|---|---|---|---|
| BrainBERT/MAE | 0.735 | 0.958 | 0.790 | 0.947 |
| **FGNO (ours)** | **0.600** | **0.965** | **0.710** | **0.954** |

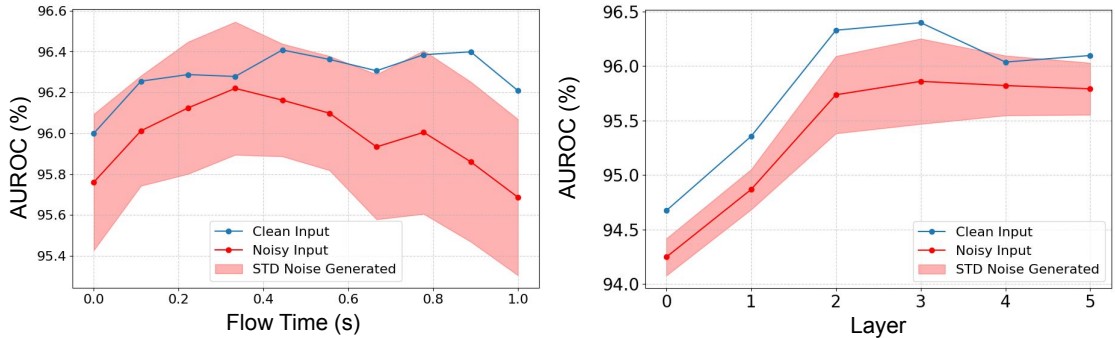

Figure 4: Sleep classification performance (DREAMT dataset, AUROC, %) comparing a "Noisy Input" against a "Clean Input" method across different model layers. **(Left)** For a representative layer, we plot performance as a function of time $s \in [0, 1]$. While both the "Clean Input" and "Noisy Input" methods exhibit the same behavioral trend, the clean input approach yields consistently higher performance. **(Right)** At the optimal time $s \approx 0.89$, the noisy method exhibits high variance over 10 runs (red-shaded region), while the clean method is deterministic and stable.

and therefore removes high-frequency spectral content. We zero-pad the unavailable frequency bins so that the resulting tensors retain the input shape expected by the pretrained FGNO. Consequently, the model is not evaluated on a new tensor discretization, but rather it is evaluated on the original grid with increasingly large regions containing no measured signal. To benchmark our function-space approach, we compare our results against MAE and Chronos, a state-of-the-art time-series foundation model that operates directly on the raw 1D data by tokenizing signal values.

As shown in Figure 5, FGNO degrades more gradually than MAE and Chronos as the downsampling factor increases. At the most aggressive setting, more than 90% of the aligned spectrogram contains no measured signal, yet FGNO retains an AUROC above 74%. These results demonstrate robustness to severe bandlimiting and to the associated zero-padding distribution shift.

**Tradeoffs of downsampling and frequency padding.** Converting inputs of different sampling resolutions into a fixed tensor shape allows a single pretrained backbone to be reused without changing its architecture. Padding the frequency axis is simple, computationally inexpensive, and preserves the alignment of the remaining low-frequency bins. However, the procedure has two limitations. First, downsampling can remove high-frequency components that may contain task-relevant information. Second, zero-padding introduces a large artificial region with no measured signal, particularly at extreme downsampling factors. Although FGNO remains robust in the experiments above, the padded values do not represent observed physiological activity and may create a distribution shift relative to the pretraining inputs. Explicit padding masks or architectures that accept variable frequency dimensions may reduce these limitations.

### 4.6 Computational Efficiency

We further evaluate the computational efficiency of FGNO, focusing on both parameter count and runtime cost. In Fig 3, FGNO achieves competitive performance across BrainTreeBank tasks with only 370K parameters, which are two orders of magnitude smaller than BrainBERT (43M) and PopT (20M). This compact architecture translates to significant efficiency gains during the probing phase. As detailed in Table 4 that compares the runtime of FGNO against the MAE/BrainBERT baseline on a single NVIDIA RTX 4090 GPU, while pre-training times are comparable to the MAE baseline, FGNO reduces the downstream adaptation time by approximately 60%. We want to note that PopT efficiency is similar to BrainBERT because it uses BrainBERT as the encoder and perform additional training objective. The speedup comes from FGNO being able to yield robust representations that require only a lightweight probing head to be trained on a frozen backbone, whereas baselines often necessitate computationally expensive full-model fine-tuning. Combined with rapid inference speeds (0.30s), FGNO offers a superior efficient solution.

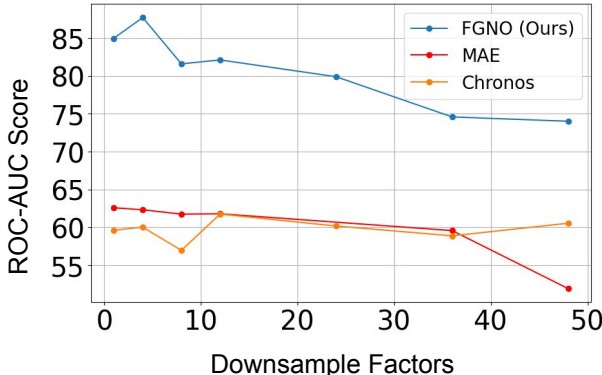

Figure 5: BrainTreeBank speech-classification performance under severe bandlimiting. FGNO is pretrained once on signals sampled at 2048 Hz and evaluated on signals downsampled by increasing factors before STFT computation. Missing high-frequency bins are zero-padded to preserve the fixed input shape. FGNO degrades more gradually than MAE and Chronos, demonstrating robustness to removed high-frequency content and padding-induced distribution shift. Because all inputs are mapped back to the original tensor shape, this experiment does not establish general discretization invariance.

Table 4: Runtime comparison between FGNO and baselines across different stages

| Model | Training | Probing/Finetuning | Inference |
|---|---|---|---|
| FGNO (Ours) | 21h 33m | 2.87 mins | 0.30s |
| BrainBERT/MAE | 19h 53m | 7.17 mins | 0.31s |

## 5 Conclusion

In this work, we presented the Flow-Guided Neural Operator (FGNO), a novel self-supervised learning framework that combines flow matching with neural operators for time-series representation learning. By embedding signals via Short-Time Fourier Transform (STFT) into time-frequency representations that can be aligned across sampling resolutions, FGNO preserves multi-scale fidelity without resampling distortions. Our framework pre-trains a single backbone model on a dataset to extract a rich hierarchy of task-specific representations. This is achieved by selecting features from an optimal network layer and a specific flow time, which acts as a continuous control for representation granularity. Furthermore, we demonstrated that using clean inputs during the probing stage, rather than the noisy inputs common in generative SSL, yields more stable and expressive features. We empirically evaluate FGNO across several biomedical time-series datasets. It demonstrates up to a 39% relative AUROC improvement over the MAE baseline in neural signal decoding on BrainTreeBank and an 18% RMSE reduction in skin temperature regression on DREAMT. Critically, FGNO shows exceptional robustness in low-data regimes, maintaining nearly full-data performance on both the SleepEDF and Epilepsy datasets with only 5% of labeled data. Its effectiveness as a neural operator is confirmed by its stable performance on BrainTreeBank across various downsampling, where baselines like MAE and Chronos degrade significantly.

The main limitation of our approach is the reliance on grid search to find the optimal $(l, s)$ pair, though it remains computationally efficient during probing. We also do not evaluate forecasting tasks, which would require a forecasting-specific prediction head and evaluation protocol beyond the discriminative probing setup considered here. Although our experiments demonstrate robustness to downsampling, they do not establish perfect resolution invariance across sampling rates; evaluating and improving such broader invariance is left for future work. Finally, our main results report point estimates from a single pretraining run per dataset, and baseline numbers on SleepEDF and Epilepsy are cited from prior work under matched dataloaders rather than re-trained in house; a systematic study of variance across seeds and subjects, with statistical significance testing, is an important direction for future work. In all, we envision FGNO as a step toward scalable, adaptable SSL, enabling transformative insights from large unlabeled time-series datasets.

**Acknowledgments**

Anima Anandkumar is supported in part by Bren endowed chair, ONR (MURI grant N00014-23-1-2654), and the AI2050 senior fellow program at Schmidt Sciences. Jiachen Yao is supported in part by the Naren and Vinita Gupta Fellowship. Jiayun Wang is supported in part by the Pritzker AI+Science initiative and Schmidt Sciences.

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

## Appendix

## A Implementation Details

**Training details** The pre-trained model is a 6-layer Transformer designed to process the output of a Short-Time Fourier Transform (STFT). The model's architecture was specifically configured to match the STFT output tensor shape: the model's input dimension of 132 corresponds to the number of frequency bins, and the sequence length of 196 corresponds to the number of time frames. Other key hyperparameters include a hidden dimension of 768, 12 attention heads, a feedforward dimension of 3072, a dropout rate of 0.1, and a learning rate of 0.0001.

**Baselines** Chronos was implemented using its Hugging Face checkpoint, extracting features with the built-in `encode` function on the same train/test windows as FGNO, though pre-trained on a broader external corpus. MAE baselines followed the BrainBERT implementation. The results of PopT are sourced from its original paper.

**Evaluation metrics** We evaluated the performance on the two downstream tasks using standard metrics. For the binary sleep classification task (awake vs. asleep), we used the Area Under the Receiver Operating Characteristic curve (AUROC). For the skin temperature regression task, we used Mean Absolute Error, Mean Squared Error (MSE), and Root Mean Squared Error (RMSE) to quantify the model's predictive accuracy.

## B Data Preprocessing and Splitting

For BrainTreeBank, we adopt the BrainBERT preprocessing pipeline for the raw time series signal data. Afterward, We segment data into non-overlapping 5-second windows, apply STFT, and treat each electrode independently. Splitting is done chronologically by subject, the same way it was done in BrainBERT.

For DREAMT, BVP and ACC signals are provided clean. We apply 64 Hz STFT and use 5-second windows.

For SleepEDF/Epilepsy, we use the official TS-TCC preprocessing pipeline to handle data preprocessing. We reuse their released dataloaders to ensure consistency in training. STFT parameters for these two datasets are the same as those used with BrainTreeBank.

Across all datasets, FGNO is pretrained on the training split only. During probing, we use held-out subjects with an 80/10/10 split for train/validation/test, respectively. Baselines use identical folds.

**Choice of STFT parameters**: The STFT window length determines the tradeoff between temporal and frequency resolution: longer windows provide finer frequency resolution but blur short-duration temporal events, whereas shorter windows improve temporal localization at the cost of coarser frequency resolution. We therefore select dataset-specific parameters based on the sampling structure and temporal scale of each dataset. For BrainTreeBank, SleepEDF, and Epilepsy, we use nperseg=400 and noverlap=350; for DREAMT, we use nperseg=64 and noverlap=48. These values are held fixed across all methods using spectrogram inputs within each dataset. We do not claim that these settings are universally optimal. The best window and overlap may depend on the downstream task—for example, sleep staging and seizure detection may favor different spectral resolutions than short-scale wearable-signal classification. A systematic evaluation of STFT parameter sensitivity is left for future work.

## C Additional Results

Here we provide the detailed performance under different layers and flow times.

Table 5: AUROC (↑) comparison between our model and MAE on DREAMT for sleep classification.

| Layer Number | FGNO (Best AUROC % @ Time) | MAE AUROC % |
|:---:|:---:|:---:|
| 1 | 94.6 @ s=1.00 | **95.8** |
| 2 | 94.6 @ s=0.89 | **95.6** |
| 3 | **96.5** @ s=0.89 | 95.7 |
| 4 | **95.9** @ s=0.67 | 95.4 |
| 5 | **96.2** @ s=0.78 | 95.5 |
| 6 | **96.4** @ s=0.89 | 95.8 |

Table 6: Best RMSE (↓) values against MAE on DREAMT for skin temperature regression task.

| Layer Number | FGNO (Best RMSE °C @ Time) | MAE RMSE °C |
|:---:|:---:|:---:|
| 1 | **0.743** @ s=0.22 | 0.790 |
| 2 | **0.691** @ s=0.33 | 0.775 |
| 3 | **0.656** @ s=0.44 | 0.735 |
| 4 | **0.625** @ s=0.33 | 0.782 |
| 5 | **0.619** @ s=0.44 | 0.738 |
| 6 | **0.600** @ s=0.56 | 0.744 |

Table 7: AUROC (↑) comparison at optimal extraction time on BrainTreeBank for speech detection.

| Layer Number | FGNO (Best AUROC % @ Time) | MAE AUROC % |
|:---:|:---:|:---:|
| 1 | **80.1** @ s=0.778 | 60.7 |
| 2 | **81.3** @ s=0.778 | 67.2 |
| 3 | **82.7** @ s=0.778 | 62.7 |
| 4 | **84.9** @ s=0.778 | 65.5 |
| 5 | **85.9** @ s=0.889 | 63.5 |
| 6 | **93.4** @ s=0.889 | 67.2 |

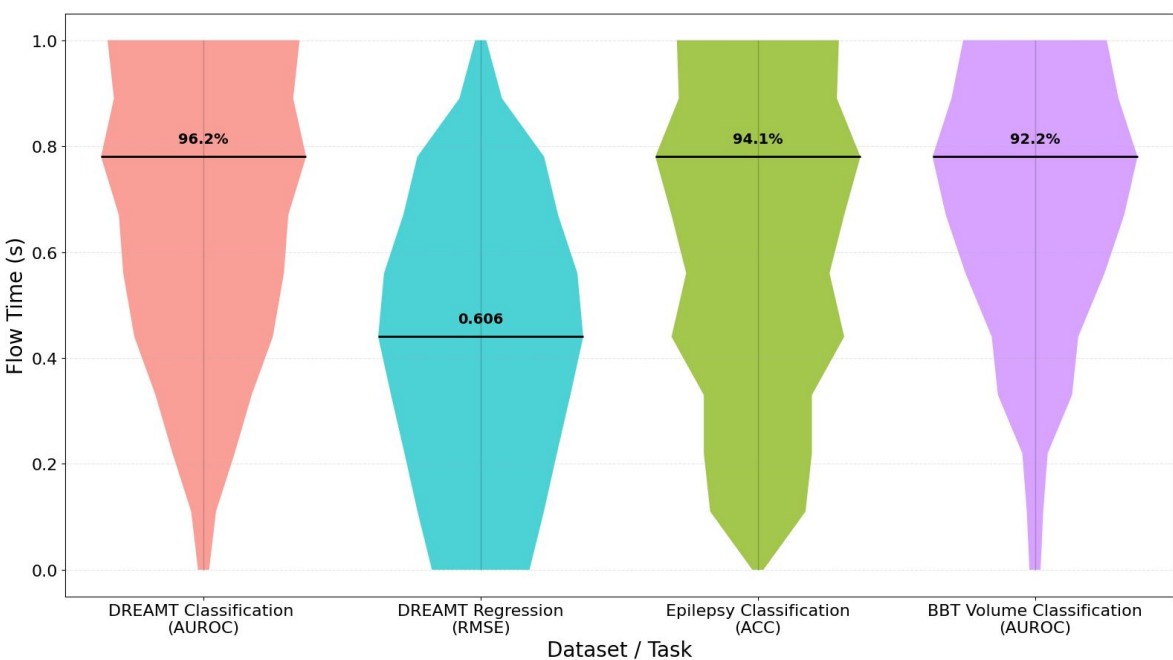

Figure 6: Task comparison across different datasets, showing that different types of tasks favor different optimal flow times $s$. The violin width at each flow time is proportional to task performance (wider is better), and the horizontal line marks the optimal flow time with its corresponding metric value. Classification tasks peak at higher flow times (lower noise), whereas the regression task peaks at intermediate flow times (higher noise).

