# OpenReview forum: "Self-Supervised Learning via Flow-Guided Neural Operator on Time-Series Data"
_TMLR — Accepted by TMLR_

### Review · Reviewer_ygKC · 2026-03-30

**Summary Of Contributions:**

This paper introduces FGNO, a self-supervised learning framework for time-series data that combines flow matching with a spectrogram specified encoder and a probing scheme. Their central idea is to treat the corruption level as a continuous control variable for representation learning, rather than relying on a fixed masking ratio as in MAE-style pretraining. The method first maps 1D signals to STFT spectrograms, then trains a time-conditioned model with a flow-matching objective, and finally probes frozen representations extracted at different layer/time pairs.

**Audience:**

Yes

**Audience Explanation:**

Yes. I believe the paper would be of interest to at least a portion of the TMLR audience, especially readers working on self-supervised learning, time series, representation learning from generative models, and robust modeling across heterogeneous temporal resolutions. The paper sits at an interesting combination, in which it draws ideas from flow matching and diffusion-style representation hierarchies and applies them to time-series SSL with a downstream evaluation workflow.

**Broader Impact Concerns:**

I do not see major broader-impact concerns specific to this work.

**Claims And Evidence:**

Yes

**Claims Explanation:**

Yes, to a reasonable extent, although I think some claims are not yet fully supported.

The paper provides a fairly clear empirical story: the proposed method improves over several baselines on multiple biomedical downstream tasks, remains competitive in low-label regimes, and includes targeted analyses on clean-vs-noisy probing and robustness to resolution changes. These experiments do support the core claim that the proposed strategy can produce useful and flexible time-series representations.

My concern is that the paper sometimes moves too fast from empirical observations to broader conceptual claims. For example, the framing around function-space learning and neural-operator-style generalization appears much stronger than what is strictly established by paper's methodology and experiments.

**Requested Changes:**

I think some claims are not yet fully supported.

1: In its method sections like section 3.1, it states that the flow dynamics operate in a finite-dimensional latent space rather than performing true function-space flow matching. However, as this paper states, it seems that it represents a function-space modeling than what is actually implemented. This makes people a little bit confused. More explanations and clarifications should be provided here.

2: I have some thinkings about the cross-resolution claim in this paper. The evaluation is interesting, but it is based on a fairly specific workflow: the downsampled signals are converted to STFTs and then zero-padded along the frequency axis to match the pretrained input dimensionality, and under extreme downsampling the paper notes that over 90% of the spectrogram contains no signal. Given this setup, I think the current experiment shows robustness under this particular evaluation workflow, but it is less clear that if it can support resolution-wide generalization. More experiments about this should be explored.

3: I also have a concern about its evaluation baseline. Chronos uses a publicly available checkpoint pretrained on a broader external corpus, while PopT results are sourced from the original paper. Since not all baselines appear to be rerun under a same experiment pattern, the fairness of some comparisons is difficult to assess.

---

> ### Author Response · Authors · 2026-04-04
> **Reply to Reviewer ygKC**
>
> We thank the reviewer for their thoughtful comments and recognizing the novelty of our approach to use flow-matching pre-training for time-series SSL. Below, we will respond to each point in detail and the changes will be reflected in our manuscript.
>
> > [R1] Clarification on function-space vs latent-space formulation
>
> We appreciate the reviewer's request for clarification. We agree that the original wording could be made more precise, and we will revise the manuscript accordingly. Specifically, we will replace the current paragraph in Section 3.1 with:
>
> Importantly, our approach is not an instance of function-space flow matching (FFM) \citep{kerrigan2023functionalflowmatching}. The inputs are time-series signals, but the flow itself is applied to encoder-derived representations rather than directly to functions. Accordingly, unlike previous work, our method does not model Gaussian random fields or use neural operator backbones for generation. The flexibility to handle variable-length or masked signals instead comes from the encoder-decoder architecture, which maps such inputs into a shared representation through padding or masking. This separation keeps the flow formulation simple while enabling flexible signal modeling.
>
> > [R2] Interpretation of cross-resolution generalization
>
> We thank the reviewer for their comment and for carefully examining the evaluation setup.
>
> We would like to first ask for clarification on what the reviewer specifically means by “support resolution-wide generalization.” In particular, it would be helpful to understand whether this refers to:
>
> - supporting both higher sampling rate and lower sampling rate in the probing process,
> - robustness to changes in the spectrogram grid-size after downsampling
>
> Based on our understanding, we believe the reviewer is pointing toward concerns around bandlimiting effects induced by downsampling and whether our current experiment truly demonstrates generalization across resolutions.
>
> To clarify our current setup:
>
> - We first downsample the raw signal in the time domain, and then compute the STFT.
> - The resulting spectrogram is zero-padded along the frequency axis to match the pretrained input dimensionality.
> - Under extreme downsampling, this results in over 90% of the spectrogram containing no signal, effectively creating a severely bandlimited input.
>
> Following the reviewer’s suggestion, we are happy to revise the manuscript by replace stronger claims about “resolution invariance” with robustness to downsampling / bandlimiting.
>
> Regardless, we would like to emphasize the practical significance of this result. In real-world time-series applications, high-frequency components are often missing, noisy, or corrupted. Therefore, the ability of FGNO to maintain strong performance (achieved better result than Chronos and MAE by large margins) even when the majority of the spectrum is empty highlights its robustness in real world setting.
>
> At the same time, we would like to emphasize the practical significance of this result. In real-world time-series settings, high-frequency components are often missing, noisy, or corrupted. The fact that FGNO maintains strong performance highlights its robustness in realistic deployment scenarios. Importantly, this robustness is learned automatically by the model, without manual tuning or auxiliary design, suggesting that it emerges naturally from the learned representation.
>
>
> > [R3] Fairness and consistency of baseline comparisons
>
> We thank the reviewer for raising this concern. We would like to clarify that comparisons with PopT are conducted only on the BrainTreeBank dataset. It is the same task as defined in BrainBERT and used in the original PopT evaluation. All methods except Chronos do not have access to privileged information.
>
> For the remaining datasets, where MAE serves as the primary baseline, we pretrain and probe BrainBERT ourselves to adhere to standard benchmarking protocols. Importantly, for all datasets where we run experiments, we maintain a consistent experimental pipeline for all models. When results are drawn from prior work, we state this explicitly. We believe this is a fair choice, as papers generally report the best available results for their own approaches.

---

### Review · Reviewer_Fhsj · 2026-04-11

**Summary Of Contributions:**

The paper proposes FGNO, a self-supervised time-series method that combines flow matching with an Short-Time Fourier Transform (STFT) based spectrogram frontend. It treats flow time s as a controllable parameter for representation learning, alongside the network layer l. The different (l,s) pairs expose different levels of abstraction. It describes clean-input probing: pretrain with noisy/interpolated inputs, but extract downstream features from clean inputs conditioned on s, avoiding stochasticity at inference. The reported empirical gains on several biomedical datasets are impressive, especially for low-label settings and robustness to downsampling. Overall, the main idea is interesting: one pretrained model yields a family of representations - from low-level patterns to high-level global features, using a single model adaptable to specific tasks.

**Additional Comments:**

The rebuttal clarified the key issues raised by this reviewer.

**Audience:**

Yes

**Audience Explanation:**

Yes, this will be of interest to the time-series prediction community as well as the biomedical community because of the choice of datasets.

**Broader Impact Concerns:**

No broader impact concern.

**Claims And Evidence:**

Yes

**Claims Explanation:**

The idea that different tasks prefer different (l,s) settings is supported by experiments. For example, sleep classification prefers higher s / lower corruption, while skin-temperature regression prefers lower-to-intermediate s. That is a good demonstration of task-dependent feature selection.  The physiological regression task depends more on global features, which align with lower to intermediate flow times (higher corruption levels), and classification tasks require more local patterns that prefer higher flow times (lower corruption), where temporal features are preserved.

Experiments also supports the claim that clean-input probing can match or slightly exceed noisy probing while removing randomness. In Figure 4, both the “Clean Input” and “Noisy Input” methods exhibit the same behavioral trend, but the clean input approach yields consistently higher performance.

**Requested Changes:**

Is the used time-conditioned Transformer on fixed-size STFT tensors truly a neural operator in the usual sense? Is function-space flow matching being performed? While the downsampling experiment is interesting, it does not by itself establish that the model is a neural operator. It shows robustness to missing high-frequency content but not necessarily true operator-level discretization invariance. Clarification on this will be helpful.

Is STFT “resolution-invariant”? Is it automatically invariant across sampling rates?

---

> ### Author Response · Authors · 2026-04-21
> **Reply to Reviewer Fhsj**
>
> We thank the reviewer for their thoughtful comments and recognizing the novelty of our approach to use flow-matching pre-training for time-series SSL. Below, we will respond to each point in detail and the changes will be reflected in our manuscript.
>
> > [R1] Clarification on Function-Space Flow Matching and Neural Operators
>
> *Reviewer Question: Is the used time-conditioned Transformer on fixed-size STFT tensors truly a neural operator in the usual sense? Is function-space flow matching being performed?*
>
> We appreciate the reviewer for asking us to clarify this point. Function-space flow matching is being performed in our framework.Standard Flow Matching typically relies on independent Gaussian noise for the base distribution. In our implementation, however, we utilize a Gaussian Process Prior (GP Prior). By using a GP, the initial "noise" state $x_0$ is not a discrete grid of independent random variables. Rather, it possesses a structured smoothness defined by a kernel (parameterized by kernel length and kernel variance). This treats the data distribution as a distribution over continuous functions. This is similar to the approach taken by Fourier Neural Operators (FNO).
>
> Furthermore, our approach to modeling data representations in the frequency domain is highly analogous to FNOs. While standard FNOs directly apply a Fourier layer to learn in the frequency domain, our framework leverages the STFT to achieve functional-style representation learning in the time-frequency domain. We will expand upon this distinction in the revised manuscript to make the function-space flow matching explicit.
>
> > [R2] Resolution Invariance vs. Severe Bandlimiting
>
> *Reviewer Question: Is STFT "resolution-invariant"? Is it automatically invariant across sampling rates? Downsampling shows robustness to missing high-frequency content but not necessarily true operator-level discretization invariance.*
>
> We thank the reviewer for pointing out this observation. Our current experiments using downsampled signals, which are converted to STFTs and zero-padded, demonstrate robustness to missing high-frequency content. To ensure our conceptual claims strictly align with our empirical evidence, we can revise the manuscript to replace the term "resolution invariance" with "robustness to severe bandlimiting."
>
> Additionally, We will clarify that STFT preprocessor does not automatically guarantee resolution invariance, but with how FGNO is designed, the model can handle severe bandlimitting in the data. Importantly, the model is trained without prior knowledge of specific downstream low-resolution tasks.
>
> This still has practical significance. In real world biomedical and time-series applications, high-frequency bands are frequently corrupted or missing. Our experiments demonstrate that the model remains highly robust even when $>90\%$ of the signal space is effectively empty. We will update the text to emphasize this practical bandlimiting robustness rather than broader operator-level invariance.

---

> > ### Comment · Reviewer_Fhsj · 2026-05-15
> > **Thank you**
> >
> > The clarifications were helpful.

---

### Review · Reviewer_SCw4 · 2026-04-30

**Summary Of Contributions:**

The paper proposes flow-guided neural operator as a model for time series data. The model combines flow matching and operator learning for learning generalizable representations for time series. This is challenging due to the heterogeneous nature of real-world data and the diversity of downstream tasks requiring, say, different sampling rates such as wearable devices collecting data at multiple frequencies. Downstream time series tasks need representations at different temporal and semantic scales and resolutions. Some downstream tasks require small time periods and others depend on large scales.
The model allows multi resolution multiscale features by training a flow matching model on spectrograms of unlabeled data. Choosing a network layer and flow time allows selection of features appropriate for a specific downstream discriminative time series  task. The STFT allows time-frequency representations for training. Flow time and network layer then become tunable features that allow selection of representations at any level in the hierarchy. The best layer and time for a task depends on the task depending on the scale of the required features.
The model learns and operators over the time-frequency spectrogram features. Such features are obtained using the Short-time fourier transform which allows extraction of both spatial and temporal features with locality properties. Multiscale features are also obtainable via diffusion and flow models and the model trains a flow matching method that allows extraction of features at multiple scales.
The model is trained in two steps: a flow matching step trains a flow matching model on the function and a feature extraction and probing phase which works on clean data to extract a representation at layer l and time s over which a probing head is trained.

The method is evaluated on a number of discriminative timeseries tasks, showing improvements in classification and robustness.

**Audience:**

Yes

**Audience Explanation:**

The method is an interesting approach for solving discriminative time series tasks with multiscale representations.

**Claims And Evidence:**

Yes

**Claims Explanation:**

The model is evaluated on wearable device data, neural responses, seizure recognition and a sleep classification dataset. The results are compared on representations obtained using MAE and Chronos. Analysis of the representations shows classification accuracy improvement with later layers on the sleep data, showing the multiscale nature of the representations.

On BrainTreeBank the method is comparable against model with domain knowledge. Finally the method also shows robustness improvement.

**Requested Changes:**

Last paragraph first page: what is missing in neural operators for which we need the stft? The paragraph is not clear about this.

For STFT the time window has to be specified. How do the results depend on this?

Sec 3.2: “To address this while preserving consistency, we extract representations”
What is meant by consistency here?

How do you test the distribution shift mitigation with the probing head? Do you test the adaptation with and without the probing head to see whether domain shift is reduced? Do you also compare with adding randomness during inference time?
How does this approach extend to forecasting tasks?
Variable length signals are handled by padding or masking into a fixed dimensional latent space. What are the pros and cons of such an approach?

typos
page 5. last paragraph condition -> conditional
sec 3.2 desiredflow -> space missing

---

> ### Author Response · Authors · 2026-05-28
>
> We thank the reviewer for their thoughtful comments and recognizing the novelty of our approach. Our respond below will be reflected in our manuscript.
>
> > [R1] Clarification for why we need STFT in this neural operator framework
>
> Our intention was not to claim that neural operators cannot model raw time-series signals. Rather, STFT is motivated by the fact that many biomedical tasks depend on localized time-frequency structure. While neural operators applied to raw 1D signals must implicitly learn frequency-localized structure, STFT explicitly exposes both temporal locality and frequency content — important for EEG, BVP, ACC, and sleep-stage recordings. We will revise the manuscript to clarify that STFT provides a task-aligned representation without implying Fourier neural operators are fundamentally inapplicable.
>
> > [R2] Clarification on STFT time window parameters
>
> We use dataset-specific STFT parameters matched to the temporal resolution of each dataset. For BrainTreeBank, SleepEDF, and Epilepsy: nperseg=400, noverlap=350. For DREAMT: nperseg=64, noverlap=48. The optimal choice depends on the downstream task, e.g. seizure detection or sleep staging may benefit from different frequency resolution than physiological signal classification.
>
> > [R3] Consistency clarification
>
> We thank the reviewer for identifying this unclear phrasing. “Consistency” is probably a bad word choice. We meant that the representation extraction procedure should be performed under the same clean input condition across all downstream tasks and all candidate layer/time pairs.
>
> During flow-matching pretraining, the model observes noisy interpolated states at different flow times. However, during downstream probing, injecting noise into the input can introduce randomness into the extracted representation and make downstream evaluation unstable. Therefore, we extract representations from clean data while varying the selected layer and flow time. This allows us to compare candidate representations in a controlled way without stochastic variation from corrupted inputs.
>
> We will revise this sentence to the following:
>
> “To avoid stochastic variation during downstream evaluation, we extract all candidate representations from clean inputs while varying only the selected network layer and flow time.”
>
> > [R4] Addressing distribution shift mitigation
>
> We thank the reviewer for this helpful question. Our current use of the probing head is primarily intended to evaluate the quality and adaptability of frozen FGNO representations for downstream tasks. The probing head is not meant to fully eliminate domain shift by itself. Rather, it provides a lightweight supervised adaptation layer on top of pretrained representations.
>
> In the current manuscript, we evaluate this by freezing the pretrained FGNO backbone and training only a simple downstream probing head. This tests whether the learned representation transfers to downstream datasets and tasks without requiring full model fine-tuning. The current experiments show that the representations are useful under downstream task transfer and reduced-label settings, but they do not directly isolate the probing head as a mechanism for reducing domain shift.
>
> > [R5] Clarification on adding randomness during inference time
>
> We thank the reviewer for this suggestion. In our current experiments, we do not add randomness during inference time. Our downstream probing uses clean inputs for representation extraction because we found that this produces more stable and reliable representations through our ablation study.
>
> The motivation is that although the model is trained through noisy flow-matching trajectories, inference-time noise can introduce unnecessary stochasticity into the extracted features. Since downstream classification tasks require deterministic and reproducible features, we use clean inputs during probing.
>
> > [R6] Using FGNO for forecasting tasks
>
> We thank the reviewer for this important question. The current paper focuses on discriminative downstream tasks, such as classification and regression from learned representations. Forecasting is an interesting extension, but it requires a different downstream evaluation setup.
>
> > [R7] Pros and cons for padding spectrograms
>
> We thank the reviewer for raising this point. Variable length signals are handled by downsampling, converting to a spectrogram, and padding missing frequency bins to match the original input dimension.
>
> The main advantage is that this keeps STFT input dimensions consistent across samples, allowing the same pretrained model to be reused across downstream tasks. The primary limitations are that downsampling may discard informative high-frequency content, and zero-padding introduces artificial values in the frequency dimension that carry no signal information. We will add a brief discussion of these tradeoffs in the limitations section.
>
> [R8] We will correct both typos. Thank you.

---

### Decision · Action_Editor_GhJm · 2026-06-05

**Recommendation:** Accept as is

**Audience:**

Yes

**Audience Explanation:**

Modeling temporal systems is an important problem in machine learning. This paper develops techniques for self-supervised learning, which is a popular approach.

**Claims And Evidence:**

Yes

**Claims Explanation:**

This paper proposes FGNO, a framwork for self supervised learning in time series using flow matching and short-time fourier transforms. The reviewers agreed that the idea is interesting, tackling an important problem: in time series, downstream tasks can require very heterogeneous features, so yielding a family of representations, a single model can be better adaptable. The main concerns were that the proposed method is largely empirical and there was a question on the fairness of the same baselines which I think the author's addressed well. Overall, all reviewers agree that this is a good contribution.